# Describing Trends in Maternal Mortality in the State of São Paulo, Brazil, from 2009 to 2019

**DOI:** 10.3390/healthcare11182522

**Published:** 2023-09-12

**Authors:** Luciane Amorim da Silva Bueno, Mariane Albuquerque Lima Ribeiro, Camila Bertini Martins, Luiz Carlos de Abreu, Alvaro Dantas de Almeida, João Antonio Correa

**Affiliations:** 1Programa de Pós-Graduação em Ciências da Saúde, Centro Universitário FMABC, São Paulo 09060-870, Brazil; luciane.bueno@fmabc.net (L.A.d.S.B.); joao.correa@fmabc.net (J.A.C.); 2Laboratório de Delineamento de Estudo e Escrita Científica, Centro Universitário FMABC, São Paulo 09060-870, Brazil; 3Centro de Ciências da Saúde e do Desporto, Universidade Federal do Acre, Rio Branco 69917-400, Brazil; 4Departamento de Medicina Preventiva, Escola Paulista de Medicina, Universidade Federal de São Paulo, São Paulo 04023-062, Brazil; cb.martins@unifesp.br; 5Departamento em Educação Integrada em Saúde, Universidade Federal Espírito Santo, Vitória 29027-502, Brazil; 6Programa de Pós-Graduação em Ciências Médicas, Faculdade de Medicina da Universidade São Paulo, São Paulo 01246-903, Brazil; alvaro.dantas@usp.br

**Keywords:** maternal death, maternal mortality, basic cause of death, maternal health service, epidemiology

## Abstract

Background: Maternal mortality is a significant public health concern, with varying impacts across different regions in Brazil, particularly affecting women from lower-income social classes with limited access to social resources. The aim of this study is to describe the trends in maternal mortality in São Paulo, Brazil, from 2009 to 2019. Materials and Methods: This study employed an ecological approach utilizing a time-series design to examine maternal deaths. Secondary data from the Mortality Information System (SIM) and the Live Births Information System (SINASC) from 2009 to 2019 were utilized. The analysis included all maternal deaths among women aged 10 to 49 years residing in the state of São Paulo. Time-series data for maternal mortality ratios were constructed for the seven regions within São Paulo State. Joinpoint regression analysis was applied to characterize the maternal mortality ratio. The study estimated the annual percentage variation, the average annual percentage variation, and their respective 95% confidence intervals. Results: In São Paulo, a total of 3075 maternal deaths were reported, resulting in a mortality ratio of 45.9 deaths per 100,000 live births. The leading causes of maternal death were eclampsia (7.13%), gestational hypertension (6.09%), and postpartum hemorrhage (5.89%). The analysis of the annual percentage change in the maternal mortality ratio for São Paulo State and its six clusters showed stationarity. Conclusions: The assessment of the maternal mortality ratio in the state of São Paulo, Greater São Paulo, and Baixada Santista revealed an increase in the maternal death ratio over the studied period.

## 1. Introduction

According to the International Statistical Classification of Diseases and Related Health Problems (ICD-10) and the World Health Organization (WHO), maternal mortality is defined as “The death of a woman during or up to 42 days after termination of pregnancy, regardless of the duration and location of the pregnancy, for any cause related to or aggravated by the pregnancy or its management, but not due to accidental or incidental causes” [1,2].

Maternal death remains a persistent public health problem worldwide, particularly prevalent in less developed regions. Alarmingly, women from low- and middle-income countries account for a staggering 94% of global maternal deaths [3].

In Brazil, there has been a reduction in the indicator in recent decades. The maternal mortality ratio (MMR) decreased from 140 maternal deaths per 100,000 live births in 1990 to 65 per 100,000 live births in 2015, which still represents a high value. Despite this improvement, it failed to achieve the objective of the United Nations (UN) Millennium Development Goals agreement [4,5,6]. The goal of the fifth Millennium Development Goal (MDG), “Improving Maternal Health”, for Brazil was to present an MMR equal to or less than 35 deaths per 100,000 live births (LB) by 2015 [7].

Considering the possibility of accelerating the decline of maternal mortality, there is a pressing need to significantly reduce this issue. Between 2016 and 2030, one of the most significant global initiatives was the inclusion of maternal mortality in the Sustainable Development Goals (SDGs) and the Global Strategy for Women’s, Children’s, and Adolescents’ Health. This initiative includes the goal of reducing the global maternal mortality ratio to less than 70 per 100,000 live births, with no country having a maternal mortality ratio greater than twice the global average [8].

Given the geographic and socioeconomic diversity in the administrative regions of the state of São Paulo, different profiles related to this indicator are observed. The aim is to assess the reasons and records of maternal mortality to describe the potential risks of pregnancy, childbirth, and the puerperium. Furthermore, it is essential to examine the health conditions of women in vulnerable situations and their social and cultural barriers to implementing more effective and immediately applicable health strategies. This analysis aims to promote prevention and provide comprehensive and humanized assisted care for these women, based on evidence-based best practices, considering their specific needs and challenges.

Therefore, the objective of this study is to describe trends in maternal mortality in Sao Paulo, Brazil, from 2009 to 2019.

## 2. Materials and Methods

### 2.1. Study Design

This study adopts an ecological approach with a time-series design, using secondary data. It includes all maternal deaths among women aged 10 to 49 years who resided in the state of São Paulo, Brazil, from 2009 to 2019.

### 2.2. Study Location

Data collection was carried out using available data from the Department of Informatics of the Unified Health System (SUS) through DATASUS (Department of Informatics of the Unified Health System—www2.datasus.gov.br (accessed on 1 August 2020)). The study utilized information from two key sources: the Mortality Information System (SIM—Sistema de Informação de Mortalidade) and the Live Birth Information System (SINASC—Sistema de Informação de Nascidos Vivos) in the state of São Paulo, Brazil.

The SINASC serves as a database populated by a standardized instrument known as the live birth certificate (DN), while the SIM is completed through the death certificate. It is worth noting that the death certificate is exclusively filled out by the attending physician, and the information concerning the underlying cause of death is extracted from the International Classification of Diseases (ICD 10/11), specifically in line “d” of this document.

### 2.3. Study Population

This study focused on recording maternal deaths among residents of the state of São Paulo from 2009 to 2019, as reported in the Mortality Information System (SIM). Maternal death was defined as “the death of a woman during pregnancy or within 42 days after the end of pregnancy, regardless of the duration or location of the pregnancy, due to any cause related to or aggravated by the pregnancy or its management, but not due to accidental or incidental causes”. This definition specifically encompassed women aged 10 to 49 years [9,10,11].

It is important to mention that, internationally, women of reproductive age are typically considered to be between 15 and 49 years old. However, in Brazil, the definition of the fertile age range is more comprehensive, spanning from 10 to 49 years old. This definition was adopted based on the insights provided by maternal death committees, vital registration statistics, and medical procedure records, which have revealed occurrences of pregnancy in women under the age of 15 [9,10].

### 2.4. Inclusion and Exclusion Criteria

This study included all maternal deaths of women aged between 10 and 49 years, residing in the state of São Paulo, Brazil, during the period from 2009 to 2019. Maternal deaths were defined based on the specific ICD-10 codes under Chapter XV titled “Pregnancy, Childbirth, and Puerperium”, which encompassed codes O00 to O99, excluding deaths outside the 42-day puerperium period (codes O96 and O97). Additionally, deaths caused by HIV (B20-B24), malignant or invasive hydatidiform mole (D39.2), or postpartum pituitary necrosis (E23.0) were considered, provided that the woman was pregnant at the time of death or had been pregnant up to one year before the occurrence of death. In cases where there was inconsistency between the reported maternal cause and the timing of death (occurring during pregnancy, childbirth, or abortion; during the puerperium up to 42 days; during the puerperium from 43 days to 1 year; or outside these periods), priority was given to information on the cause to determine whether the death was maternal or not [10,11].

Excluded from the study were death records due to other causes not relevant to the pregnancy–puerperal process, as well as death records occurring more than one year after delivery.

### 2.5. Data Collection

The data used in this study were obtained through the file transfer service provided by the Department of Informatics of the Unified Health System (DATASUS) via their official website: http://datasus.saude.gov.br/ (accessed on 1 August 2020). This service facilitates the systematic registration of data on mortality (Vital Statistics—Mortality and Live Births).

To ensure data accuracy, two independent researchers were responsible for collecting and verifying the information, aiming to identify any potential discrepancies. The data querying process involved the utilization of specialized programs called TABNET and TABWIN. These tabulators were specifically developed to enable swift tabulation of DBF (dBase) and CSV (comma-separated values) files, thus expediting the data analysis process for the study.

### 2.6. Instruments for Data Extraction

The data extracted from TABNET were obtained through the use of a form for data entry.

### 2.7. Study Variables

The maternal deaths were described using the following variables: mother’s age, skin color/race, education, marital status, place of occurrence, type of cause, and year of death.

### 2.8. Statistical Analysis

The data underwent descriptive analysis to characterize the population according to sociodemographic variables. For categorical variables, the distribution of absolute and relative frequencies was utilized.

Additionally, the gross maternal mortality ratio was computed, considering both the geographic distribution (cluster) and each year from 2009 to 2019. This ratio was expressed per 100,000 live births [10,11].

The formula or method used to calculate the maternal mortality ratio was as follows:$$\frac{\begin{array}{c} The\;number\;of\;deaths\;of\;resident\;women,\;due\;to\;causes\;\\ linked\;to\;pregnacy,\;childbirth\;and\;pueperim \end{array}}{Number\;of\;live\;births\;to\;resident\;mothers}\times 100,000$$

The territorial clusters of São Paulo State were determined based on maternal addresses, and they were as follows: Grande São Paulo (GSP), the metropolitan region of Baixada Santista (BSMR), the administrative region of Taubaté (TAR), Cluster Centro Sul (comprising the administrative regions of Sorocaba and Registro—CSC), the Cluster Campinas region (including the administrative regions of Campinas, Piracicaba, and São João da Boa Vista—CRC), Cluster Centro Norte (encompassing the administrative regions of Bauru, Araraquara, Ribeirão Preto, and Franca—CNC), Cluster Noroeste (comprising Marília, Presidente Prudente, Araçatuba, and the administrative regions of São José do Rio Preto and Barretos—NWC), and the state of São Paulo as a whole (Figure 1).

These groupings were established based on similar territorial characteristics and the geographical delimitation of administrative areas [12], as previously stipulated by government agencies, utilized by DATASUS, and adapted by Calderón et al. [13]. This approach ensured an adequate number of cases and stability in the analysis.

For the current analysis, time series were constructed to examine the maternal mortality ratio between the years 2009 and 2019 for the seven regions and the state of São Paulo. We employed Joinpoint regression to identify significant changes in trends during the observed period [14]. The independent variable was time (year). We assumed a constant error variance and estimated first-order autocorrelation from the data. Additionally, we applied log-transformation to the dependent variable. The maximum number of joinpoints was predefined based on the number of data points [15]. Since we analyzed maternal mortality over an 11-year period, the maximum number of joinpoints was limited to one.

The annual percentage change (APC), average annual percentage change (AAPC), and their respective 95% confidence intervals were estimated. The significance level used for statistical analysis was 5%. The statistical analyses were conducted using the Joinpoint Regression Program, Version 4.9.0.1—February 2022 (Statistical Methodology and Applications Branch, Surveillance Research Program, National Cancer Institute) [15].

### 2.9. Ethical and Legal Aspects of Research

As the data used in this study were obtained from the public domain, there was no requirement to submit the work to the National Research Ethics Commission (CONEP) or undergo analysis by the Research Ethics Committee (CEP) system.

## 3. Results

In the state of São Paulo, a total of 3075 maternal deaths were reported between 2009 and 2019, resulting in a mortality rate of 45.9 deaths per 100,000 live births. The absolute and relative frequencies of maternal deaths were higher among the following variables: women aged 20 to 29 years (37.3%) and 30 to 39 years (43.93%), individuals of white ethnicity (56.22%) and mixed race (32.35%), those with 8 to 11 years of education (40%), unmarried individuals (50.37%), and cases where the place of death was a hospital (92.81%) (Table 1).

In relation to the obstetric causes of maternal death, a total of 3075 deaths were analyzed. Despite this, in three cases, it was not possible to determine the specific causes after investigation, resulting in a total of 3072 maternal deaths for this analysis (Table 2).

As for the classification of these deaths, the most frequent were direct causes, accounting for 61% (1874) of the sample. Among these, the most prominent were eclampsia (219), gestational hypertension (187), postpartum hemorrhage (181), obstetric embolism (126), puerperal infection (121), and unspecified abortion (NE) (115) (Table 2).

Indirect obstetric causes accounted for 36.3% (1115) of the deaths, categorized as other severe complications of childbirth and the puerperium (949), maternal parasitic infectious diseases that become complicated during pregnancy, childbirth, and the puerperium (79), pre-existing hypertension complicating pregnancy, childbirth, and the puerperium (66), and diabetes mellitus in pregnancy (19).

In the descriptive analysis of maternal deaths across the seven conglomerates in the state of São Paulo, the highest number of deaths was concentrated in the Greater São Paulo (GSP) conglomerate, with 1645 reported deaths, followed by the Campinas Region Cluster (CRC) with 395 deaths, and the Cluster Centro Norte (CNC) with 267 deaths (Table 3).

Regarding the types of obstetric causes of maternal death by conglomerate, the direct ones accounted for the highest number of deaths in all seven regions. Eclampsia (O15) was the most frequent direct cause of death in the GSP conglomerate, with 121 deaths, followed by CRC with 27 deaths and CNC with 22 deaths. Gestational hypertension (O14) was the most common direct cause of death in GSP with 90 deaths, CRC with 27 deaths, and CNC with 21 deaths. Postpartum hemorrhage (O72) was also significant in GSP with 99 deaths, CRC with 27 deaths, and CNC with 23 deaths (Table 3).

The most frequent indirect causes in all clusters were O99 (other diseases of the mother, classified elsewhere, but which complicate pregnancy, childbirth, and the puerperium), O98 (other maternal parasitic diseases of the mother that become complicated during pregnancy, childbirth, and the puerperium), O10 (pre-existing hypertension complicating pregnancy, childbirth, and the puerperium), and O24 (diabetes mellitus in pregnancy). Unknown obstetric causes had more significant numbers in the GSP and ACR conglomerates in relation to the other regions (Table 3).

The APC and AAPC of the maternal mortality ratio remained stable in most district conglomerates in the state of São Paulo from 2009 to 2019 (Table 4). However, the Cluster Centro Norte (APC = 8.9; 95% CI = (2.1; 16.1)) and Cluster Noroeste (APC = 5.5; 95% CI = (0.4; 10.9)) exhibited an increasing trend during the analyzed period. Additionally, Grande São Paulo (APC = 5.0; 95% CI = (2.6; 7.4)) and the state of São Paulo (APC = 4.8; 95% CI = (0.9; 8.8)) showed an increasing trend from 2011 to 2019.

In Figure 2, the trend of the maternal mortality ratio in São Paulo State is depicted according to the district conglomerates. The time series demonstrate a similar pattern, with Baixada Santista showing greater variability.

## 4. Discussion

The maternal mortality ratio (MMR) in the state of São Paulo experienced a reduction from 55.47 in 2009 to 48.53 in 2019, indicating a decline of 12.51%. A total of 3075 deaths were reported during this period. Maternal deaths were more prevalent among women aged 30 to 39 years (43.93%), white women (56.22%), and mixed-race women (32.35%), with 40% having completed education from 8 to 11 years. Additionally, 50.37% of the cases were unmarried, and 92.81% of deaths occurred in hospitals. The Greater São Paulo conglomerate (GSP) had the highest number of deaths, with 1645 reported cases, followed by the Campinas Region Cluster (CRC) with 395 deaths, and the Cluster Centro Norte (CNC) with 267 deaths. The MMR trends in the clusters and the state of São Paulo remained stable from 2009 to 2019.

Maternal obstetric deaths can be classified into two categories: direct and indirect. Direct deaths occur due to obstetric complications related to pregnancy, childbirth, and the puerperium, resulting from interventions, negligent omissions, incorrect treatment, or a sequence of events stemming from any of these circumstances (e.g., hemorrhage, puerperal infection, hypertension, thromboembolism, anesthetic accident) [16].

On the other hand, indirect maternal deaths arise from pre-existing or intercurrent diseases that develop during pregnancy and are not directly caused by obstetric factors. These deaths are exacerbated by the physiological effects of pregnancy, such as heart disease, collagen disorders, and other chronic illnesses [16]. Furthermore, they constitute the primary cause of maternal mortality in developing countries.

The current study highlights that the predominant causes of maternal death are of direct obstetric origin. Among these causes, arterial hypertension during pregnancy stands out as the most prevalent. Despite a decrease in its occurrence in the state of São Paulo, it still accounts for approximately 16% of all maternal deaths.

The most prevalent indirect causes in all conglomerates were as follows: other diseases of the mother, classified elsewhere, but which complicate pregnancy, childbirth, and the puerperium; other maternal parasitic diseases of the mother that are complications during pregnancy, childbirth, and the puerperium; pre-existing hypertension complicating pregnancy, childbirth, and the puerperium; and diabetes mellitus in pregnancy. Additionally, unknown obstetric causes were more prevalent in the Greater São Paulo and regional conglomerates compared to other locations.

In an extensive systematic review encompassing 180 countries, a noticeable decline in global maternal mortality was observed, attributed to multiple contributing factors. Among these factors was the reduction in the world fertility ratio, which decreased from 3.70 in 1980 to 2.56 in 2008, indicating stable global birth rates despite an increasing number of women of reproductive age. Additionally, the increase in per capita income, improvement in women’s educational levels, and enhanced quality of prenatal and labor care also played a significant role in this positive trend [17].

Maternal deaths in developing countries exhibit diverse causes depending on the region. However, the predominant proportion of maternal deaths is associated with direct obstetric causes. A comprehensive systematic review of maternal causes of death revealed that hemorrhages prevail in Africa and Asia, while hypertensive diseases, obstructed births, and abortion complications are more common in Latin America and the Caribbean. In South Africa, a strong association between maternal mortality and HIV/AIDS is evident [18].

In these developing countries, identifying indirect causes of maternal death can be challenging due to diagnostic difficulties, a lack of patient-reported pre-existing diseases, or limited awareness of such causes [17].

In Brazil, the government is committed to reducing maternal mortality and recognizes the crucial role of maternal death surveillance. Consequently, maternal mortality committees have been established and strengthened at the national, regional, state, municipal, and hospital levels [10,19].

Brazil has made significant progress in reducing maternal mortality, with a decline from 141 maternal deaths per 100,000 live births in 1990 to 64.8 per 100,000 in 2011. Notably, there was a remarkable 67% decline in maternal deaths from direct causes during this period, from 126 to 43 maternal deaths per 100,000 live births, indicating a promising 5.1% annual decline [20].

Data from the Ministry of Health indicate that the maternal mortality ratio in the national average, 64.8 per 100 thousand, in the same year, the situation in São Paulo also exceeded 35 deaths per 100 thousand people wanted. In addition, there is an unjustifiable aggravation of the numbers concerning 2003, when 32.4 deaths per 100 thousand were recorded. Still in southeastern Brazil, the State of Rio had a maternal mortality ratio of 74.3 per 100 thousand, considerably above the national average. On the other hand, Santa Ca-tarina reduced maternal death to 25.2 per 100 thousand, an index that meets the goal of the fifth MDG [6,8,21].

The Maternal Mortality Ratio in the state of São Paulo showed a slight reduction be-tween 2000 and 2008, compared to the values of the previous decade. However, there was an important increase in 2009, which is attributed, in the situation analysis of the State Plan mentioned, to the severity of the H1N1 Influenza A epidemic for pregnant women, not only because of the direct deaths of pregnant women due to Influenza but also in-crease in the lethality of other obstetric emergencies that require intensive care, the sup-ply of which at the time was dramatically compromised by cases of the epidemic [22].

Despite this progress, data from the Ministry of Health reveal that both the national average and the situation in São Paulo still exceed the UN-proposed goal of 30 deaths per 100,000 live births. In 2018, the Brazilian maternal mortality ratio was 59.1 deaths per 100,000 live births [6]. Although the state of São Paulo has a lower maternal mortality ratio than the national average, it still remains relatively high compared to developed countries, particularly in certain regions of the state [22].

Unsafe abortion significantly impacts maternal mortality in developing countries [23]. This issue may be further exacerbated by underreporting of maternal deaths related to abortion complications, influenced by cultural norms and legal implications.

Maternal mortality serves as a reflection of the sociopolitical–cultural landscape of society, just as the infant mortality rate indicates the overall health status of the population and is closely related to the availability and quality of existing healthcare resources [16].

The increasing trend of pregnancy in women of advanced age results from both a conscious decision to delay pregnancy and the easy access to assisted fertilization techniques. However, this reality has contributed to an upsurge in maternal mortality among this group of women [24].

Research generally supports the notion that maternal deaths occur more frequently in unmarried women [25]. These data could indicate a sense of vulnerability in motherhood contributing to death, as well as a potential conflation between marital status and cohabitation status [26].

Cesarean delivery has been associated with an increased risk of maternal death compared to vaginal delivery. This elevated risk is linked to complications such as thromboembolism, puerperal infection, and anesthetic issues [27].

Other factors are also considered to exacerbate the risk of maternal death, including an interbirth interval of less than two years, maternal malnutrition, obesity, and delayed initiation of prenatal care after the 24th week. Additionally, factors like overcrowded hospitals, limited access to health services, inadequate professional qualifications in care, and delays in diagnosis (and, consequently, proper treatment) can also contribute to maternal death [28].

Implementing measures such as the creation of the “Rede Cegonha”, which aims to establish a care network encompassing reproductive planning and humanized attention to pregnancy, childbirth, and the puerperium, can play a crucial role in addressing this issue. The introduction of a care pathway for pregnant women and mothers within SUS-SP is another effective measure that positively impacts the reduction in maternal deaths.

Guaranteeing the connection of pregnant women to reference units and ensuring safe transportation, proper risk classification of pregnant women (e.g., identifying pre-eclampsia), and providing continuous educational opportunities for doctors and nurses in prenatal care, childbirth, and obstetric emergencies, among other initiatives, can significantly improve the quality of care provided to women.

A comprehensive literature review highlighted the importance of conducting prenatal care to identify potential risks, providing nutritional support to pregnant women, treating diseases, and implementing a maternal immunization program to reduce obstetric risks [29].

Moreover, reinforcing the maternal death surveillance committees has proven to be a pivotal policy. These committees serve the dual purpose of understanding the true scope of the issue in each region and engaging in discussions with health services responsible for pregnant women’s care, as well as the basic network that administers prenatal care. Analyzing the type of care provided to pregnant women, identifying shortcomings, and proposing effective solutions are vital steps towards expediting the reduction in maternal mortality indicators [30].

Prevention measures for maternal deaths are the review of cases by the Maternal Death Committees, which must adopt an active and participatory stance. Everyone in-volved in the woman’s death should be questioned to identify the factors involved in a death. The case must be studied by qualified people committed to educating and not pun-ishing. The Committee’s objective is to analyze all the variables involved and propose in-terventions to prevent a new death [10,31].

Nonetheless, certain limitations of the present study warrant acknowledgment. Identifying cases of maternal death can be complex due to inadequate completion of death certificates and substantial underreporting of such deaths [30]. The results are based on secondary data from health information systems and are subject to the quality of the information recorded. Accuracy is contingent upon the proper completion of death and live birth certificates, which are then entered into the health information systems’ databases.

As this study is of an ecological nature, it inherently bears limitations regarding the establishment of causality concerning deaths. It is not feasible to definitively define such causal relationships. Nevertheless, the primary aim of this study is to describe the potential causes of deaths, shedding light on the trends of maternal mortality. This estimation encompasses not only the risks associated with pregnancy, childbirth, and the postpartum period, but also serves as an indicator of the overall health, social, and economic status of women.

Another important aspect pertains to the challenge of obtaining an accurate maternal mortality ratio, especially due to the confusion arising from incorrect classification between maternal deaths and those occurring during pregnancy. The latter category may include deaths from infectious or non-communicable diseases, as well as from external causes that are not considered maternal [4,32].

Additionally, an important issue concerns maternal deaths resulting from induced abortions in countries where abortion is illegal [33]. The illegality of abortion does not prevent such procedures from being performed, leading to the utilization of unsafe practices. When these practices result in death, the fatalities may not always be included in maternal mortality statistics [34].

Maternal mortality remains a global epidemic affecting developing countries, particularly women in less privileged economic circumstances. To curb these deaths, proposed measures include comprehensive family planning to prevent unwanted pregnancies, adequate prenatal care, a qualified team to handle obstetric emergencies, safe abortion procedures, and postpartum surveillance [35,36].

Obtaining information on maternal mortality levels and trends is essential not only for estimating risks in pregnancy and childbirth, but also for understanding women’s overall health and, by extension, their social and economic status. Estimating the magnitude of the problem is a critical first step in identifying its root causes and proposing effective improvements.

Accurately gauging the scale of this issue constitutes a pivotal initial step in identifying its underlying causes and proposing potential enhancements. From an epidemiological standpoint, this study plays a vital role in guiding the formulation and implementation of public policies directed towards women’s health, as well as enhancing the education and training of healthcare professionals. Such measures have the potential to significantly improve care during the pregnancy–puerperal cycle, family planning, and the registration and investigation of maternal deaths.

Moreover, the selection of the study period bears considerable importance for future research, facilitating comparisons with preceding and subsequent periods. This allows for a thorough analysis of the impact of public policies and the COVID-19 pandemic on maternal mortality trends.

## 5. Conclusions

The evaluation of the maternal mortality ratio (MMR) in the state of São Paulo, Greater São Paulo, and Baixada Santista revealed an increase in the maternal death ratio. Conversely, in the other conglomerates, the MMR remained stable. The occurrence of maternal deaths in the state was predominantly observed among white and mixed-race women aged 20 to 39 years, with 8 to 11 years of education, unmarried marital status, and hospitals as the place of death. Eclampsia, gestational hypertension, and postpartum hemorrhage were identified as the leading causes of maternal death.

In recent years, notable progress has been achieved in the implementation of public policies focused on women’s health. However, despite these efforts, the interventions proposed thus far have not fully addressed the specific needs of women or significantly impacted the reduction in the maternal mortality ratio in the last decade in the state of São Paulo. This indicator continues to serve as a crucial measure of health and quality of life for this population.

## Figures and Tables

**Figure 1 healthcare-11-02522-f001:**
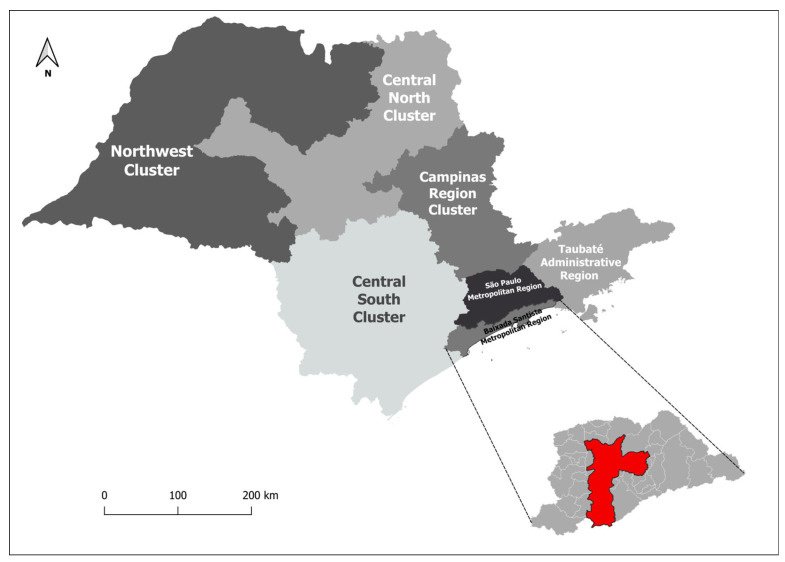
São Paulo State cluster map.

**Figure 2 healthcare-11-02522-f002:**
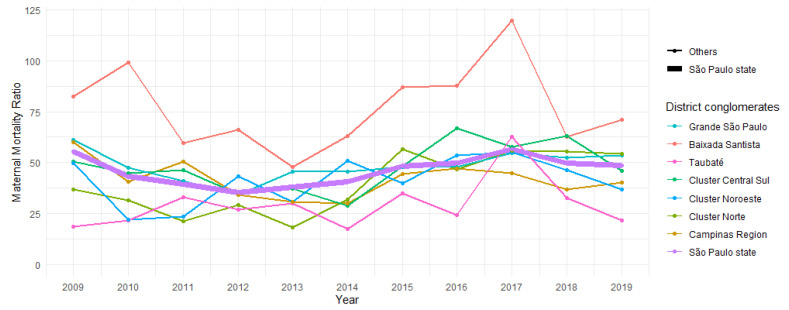
Trends of maternal mortality ratio in the district conglomerates of the state of São Paulo, Brazil, in the period from 2009 to 2019.

**Table 1 healthcare-11-02522-t001:** Frequency of maternal deaths and maternal mortality ratio in the state of São Paulo from 2009 to 2019.

Characteristics	Number of Maternal Deaths	Number of Live Births	%	Maternal Mortality Ratio *
**Age group**				
10 to 14 years	11	33,482	0.35	32.85
15 to 19 years	293	882,005	9.52	332.19
20 to 29 years	1140	3,251,041	37.03	35.06
30 to 39 years	1351	2,321,948	43.93	58.18
40 to 49 years	280	210,510	9.10	133.01
**Skin color/race**				
White	1729	4,142,023	56.22	41.74
Black	260	307,831	8.45	84.46
Yellow	11	35,251	0.35	31.20
Mixed	995	2,123,393	32.35	46.85
Indigenous	0	9712	0	0
Ignored	80	80,776	2.60	99.03
**Education**				
None	16	12,225	0.52	130.87
1–3 years	193	94,529	6.27	204.17
4–7 years	693	838,054	22.53	82.69
8–11 years	1230	4,206,867	40	29.23
12 years or more	380	1,507,636	12.36	25.20
Ignored	563	39,675	18.30	1419.02
**Marital status**				
Single	1549	2,901,981	50.37	53.37
Married	1050	2,744,176	34.14	38.26
Widow	22	12,358	0.71	178.02
Judicially separated	100	117,130	3.25	85.37
Other	247	778,756	8.03	31.71
Ignored	107	48,699	3.4	219.71
**Place of occurrence**				
Hospital	2854	6,650,684	92.81	42.91
Another health facility	92	25,246	2.99	364.41
Home	90	17,962	2.92	501.05
Public highway	5	-	0.16	-
Other	34	4943	1.10	687.84
Ignored	0	151	0	0
**Type of obstetric cause**				
Direct obstetric	1874	-	61.00	-
Indirect obstetric	1115	-	36.30	-
Unspecified obstetric	83	-	2.70	-
**Maternal death**				
2009	332	598,421	10.76	55.47
2010	259	601,289	8.40	43.07
2011	241	610,150	7.81	39.49
2012	219	616,545	7.10	35.52
2013	233	610,836	7.55	38.14
2014	254	625,590	8.23	40.60
2015	307	633,935	9.95	48.42
2016	299	601,393	9.69	49.71
2017	346	611,735	11.22	56.56
2018	302	606,065	9.79	49.82
2019	283	583,057	9.17	48.53
**Total**	**3075**	**6,698,986**	**100**	**45.90**

Data source: * The maternal mortality ratio (MMR) was calculated using data from the Mortality Information System (SIM) and the Live Birth Information System (SINASC), both available at the Informatics Department of the National Health System (DATASUS). The MMR represents the number of maternal deaths per 100,000 live births. Nevertheless, it was not possible to calculate the MMR for the indigenous population, due to the absence of maternal death data in the SIM system (Mortality Information System). Additionally, there is a limitation with regard to the place of occurrence in the SINASC itself. The SINASC does not provide information on the number of live births related to the obstetric cause of maternal death, making it impossible to calculate the MMR in this context. It is important to emphasize that these two systems are separate entities and are not interconnected.

**Table 2 healthcare-11-02522-t002:** Frequency of maternal deaths due to obstetric causes, by category, in the state of São Paulo, Brazil, from 2009 to 2019.

Category CID 10	N	%
**Direct Obstetric Causes**		
O15 Eclampsia	219	7.13
O14 Gestational hypertension with significant proteinuria	187	6.09
O72 Postpartum hemorrhage	181	5.89
O88 Obstetric origin embolism	126	4.10
O85 Postpartum infection	121	3.94
O90 Postpartum complication NCOP	115	3.74
O62 Abnormalities of uterine contraction	99	3.22
O06 Abortion NE	90	2.93
O45 Premature detachment of the placenta	79	2.57
O00 Ectopic pregnancy	75	2.44
O23 Infection of the genitourinary tract in pregnancy	71	2.31
O75 Other complications of labor and childbirth NCOP	62	2.02
O13 Gestational hypertension without significant proteinuria	39	1.27
O16 Maternal hypertension NE	35	1.14
O44 Placenta previa	35	1.14
Other direct obstetric causes *	340	11.07
**Indirect Obstetric Causes**		
O99 Other maternal illness COP compl grav childbirth puerp	949	30.89
O98 Inf paras mat disease COP compl grav childbirth puerp	79	2.57
O10 Hypertension pre-exist complic grav childbirth puerp	66	2.15
O24 Diabetes mellitus in pregnancy	19	0.62
O25 Desnutric in pregnancy	2	0.07
**Unspecified Obstetric Causes**		
O95 Obstetric death NE	83	2.70
**Total**	**3072**	**100.00**

Data source: * The Mortality Information System (SIM) records various other direct obstetric causes, each identified by specific codes. These include O71 for other obstetric trauma, O86 for other puerperal infections, O26 for assisting maternal other complications predominantly related to the gravid state, O03 for miscarriage, O67 for labor delivery childbirth complicated by hemorrhage, intrapartum, not elsewhere classified (NCOP), O87 for complicated venous problems in the puerperium, O4 for other disorders of membranes and amniotic fluid, O02 for other abnormal products of conception, O46 for NCOP antepartum hemorrhage, O05 for other types of abortion, O22 for complicated venous problems in pregnancy, O74 for complications of anesthesia during labor and delivery, O01 for hydatidiform mole, O43 for transfusion of the placenta, O07 for failed attempted abortion, O11 for pre-existing hypertension with proteinuria superimposed, O24 for diabetes mellitus in pregnancy, O73 for retained placenta and membranes without hemorrhage, O21 for excessive vomiting in pregnancy, O42 for premature rupture of membranes, O60 for pre-term labor, O68 for labor and childbirth complicated by fetal distress, O91 for infections of the mammary glands associated with childbirth, O04 for abortion for medical and legal reasons, O12 for edema and proteinuria of pregnancy without hypertension, O20 for early-pregnancy bleeding, O33 for assisting preterm mother, disproportionate, known or suspected, O34 for assisting preterm mother with other specified abnormalities of pelvic organs, O66 for other forms of obstructed labor, O08 for complications consequent to abortion, molar pregnancy, and ectopic pregnancy, O30 for multiple gestation, O36 for assisting preterm mother, other specified problem of fetus, known or suspected, O61 for failure to induce labor, O64 for obstruction of labor due to malposition and malpresentation of fetus, O65 for obstruction of labor due to a mother’s pelvic abnormality, and O89 for complications of administration of anesthesia during the puerperium.

**Table 3 healthcare-11-02522-t003:** Frequency of maternal deaths from obstetric causes by category of ICD 10 by clusters in the state of São Paulo, Brazil, from 2009 to 2019.

Category CID 10	GSP	%	BSMR	%	TAR	%	CSC	%	NWC	%	CNC	%	CRC	%	Total	%
**Direct Obstetric Causes**																
O15 Eclampsia	121	7.36	9	4.35	13	12.15	13	6.63	14	5.6	22	8.24	27	6.84	219	7.13
O14 Gestational hypertension w/proteinuria signif.	90	5.47	16	7.73	6	5.61	13	6.63	14	5.6	21	7.87	27	6.84	187	6.09
O72 Postpartum hemorrhage	99	6.02	10	4.83	6	5.61	6	3.06	10	4	27	10.11	23	5.82	181	5.89
O88 Obstetric orig embolism	87	5.29	4	1.93	5	4.67	4	2.04	12	4.8	4	1.50	10	2.53	126	4.10
O85 Puerperal infection	69	4.19	6	2.90	3	2.80	8	4.08	6	2.4	13	4.87	16	4.05	121	3.94
O90 Postpartum complication NCOP	61	3.71	5	2.42	3	2.80	7	3.57	11	4.4	12	4.49	16	4.05	115	3.74
O62 Abnormalities of uterine contraction	42	2.55	9	4.35	3	2.80	10	5.10	11	4.4	8	3.00	16	4.05	99	3.22
O06 Abortion NE	71	4.32	1	0.48	1	0.93	1	0.51	6	2.4	5	1.87	5	1.27	90	2.93
O45 Premature detachment of the placenta	39	2.37	6	2.90	2	1.87	11	5.61	10	4	6	2.25	5	1.27	79	2.57
O00 Ectopic pregnancy	35	2.13	6	2.90	3	2.80	5	2.55	7	2.8	7	2.62	12	3.04	75	2.44
O23 Infection of the genitourinary tract in pregnancy	27	1.64	4	1.93	4	3.74	4	2.04	7	2.8	7	2.62	18	4.56	71	2.31
O75 Other complications of labor and childbirth NCOP	22	1.34	4	1.93	3	2.80	11	5.61	7	2.8	12	4.49	3	0.76	62	2.02
O13 Gestational hypertension without proteinuria signif.	19	1.16	5	2.42	1	0.93	4	2.04	3	1.2	4	1.50	3	0.76	39	1.27
O16 Maternal hypertension NE	15	0.91	3	1.45	3	2.80	2	1.02	4	1.6	4	1.50	4	1.01	35	1.14
O44 Placenta previa	19	1.16	1	0.48	2	1.87	2	1.02	4	1.6	5	1.87	2	0.51	35	1.14
Other direct obstetric causes	154	9.36	28	13.53	15	14.02	30	15.31	30	12	33	12.36	50	12.66	340	11.07
**Indirect Obstetric Causes**																
O99 Other doenc mat COP compl grav childbirth puerp	528	32.10	76	36.71	23	21.50	52	26.53	79	31.6	54	20.22	137	34.68	949	30.89
O98 Inf paras mat disease COP compl grav childbirth puerp	38	2.31	10	4.83	4	3.74	6	3.06	8	3.2	6	2.25	7	1.77	79	2.57
O10 Hypertension pre-exist complic grav childbirth puerp	47	2.86	2	0.97	1	0.93	1	0.51	3	1.2	3	1.12	9	2.28	66	2.15
O24 Diabetes mellitus in pregnancy	13	0.79	1	0.48	3	2.80	1	0.51	0	0	1	0.37	0	0.00	19	0.62
O25 Malnutrition in pregnancy	1	0.06	0	0.00	0	0.00	0	0.00	0	0	1	0.37	0	0.00	2	0.07
**Unspecified Obstetric Causes**																
O95 Cause obstetric death NE	48	2.92	1	0.48	3	2.80	5	2.55	4	1.6	12	4.49	5	1.27	83	2.70
**Total**	**1645**	**100**	**207**	**100**	**107**	**100**	**196**	**100**	**250**	**100**	**267**	**100**	**395**	**100**	**3072**	**100**

Data source: Mortality Information System (SIM). Grande São Paulo (GSP), metropolitan region of Baixada Santista (BSMR), administrative region of Taubaté (TAR), Cluster Centro Sul (administrative regions of Sorocaba and Registro—CSC), Cluster Region Campinas (administrative regions Campinas, Piracicaba, and São João da Boa Vista—CRC), Cluster Centro Norte (administrative regions Bauru, Araraquara, Ribeirão Preto, and Franca—CNC), Cluster Noroeste (Marília, Presidente Prudente, Araçatuba, administrative regions of São José do Rio Preto and Barretos—NWC).

**Table 4 healthcare-11-02522-t004:** Point and interval estimates of the Joinpoint regression for the maternal mortality ratio in São Paulo State, Brazil, stratified by district conglomerates, 2009–2019.

District Conglomerates	Maternal Mortality Ratio			Range	APC (95% CI)	AAPC (95% CI)
	2009	2019	2009–2019			
Baixada Santista	82.6	71.2	77.0	2009–2019	0.7 (−4.9; 6.6)	0.7 (−4.9; 6.6)
Cluster Campinas Region	60.0	40.4	41.9	2009–2019	−1.5 (−5.8; 3.0)	−1.5 (−5.8; 3.0)
Cluster Centro Norte	36.7	54.4	39.8	2009–2019	8.9 (2.1; 16.1) ^↑^	8.9 (2.1; 16.1) ^↑^
Cluster Centro Sul	50.4	46.1	47.7	2009–2019	2.8 (−2.7; 8.6)	2.8 (−2.7; 8.6)
Cluster Noroeste	49.8	36.8	41.1	2009–2019	5.5 (0.4; 10.9) ^↑^	5.5 (0.4; 10.9) ^↑^
Grande São Paulo	61.2	53.5	48.3	2009–2011	−20.9 (−44.9; 13.6)	−0.8 (−6.5; 5.3)
				2011–2019	5.0 (2.6; 7.4) ^↑^	4.3 (−1.8; 10.8)
Taubaté	18.7	21.8	29.5	2009–2019	4.3 (−1.8; 10.8)	
São Paulo State	55.5	48.5	46.0	2009–2011	−17.5 (−38.6; 10.9)	−0.1 (−5.3; 5.3)
				2011–2019	4.8 (0.9; 8.8) ^↑^	

Data source: Mortality Information System (SIM) and Live Birth Information System (SINASC), available at the Informatics Department of the National Health System (DATASUS-www.datasus.gov.br (accessed on 1 August 2020)), Ministry of Health, Brazil. Abbreviations: APC: annual percentage change; AAPC: average annual percentage change; 95% CI: 95% confidence interval; ↑: increasing trend.

## Data Availability

The data used in this study were obtained through the file transfer service provided by the Department of Informatics of the Unified Health System (DATASUS) via their official website: http://datasus.saude.gov.br/ (accessed on 1 August 2020). This service facilitates the systematic registration of data on mortality (Vital Statistics—Mortality and Live Births).

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
