# Peer review of "Describing Trends in Maternal Mortality in the State of São Paulo, Brazil, from 2009 to 2019"

_healthcare, 2023, doi:10.3390/healthcare11182522_

Round 1

Reviewer 1 Report

The introduction part looks incomplete. It is missing in a number of ways. For example, no method was discussed, and no results were briefed.

Second, the reasons for the mortality due to the regions were not researched and not discussed. The research has very weak take-home message for the reader to link it for the actions.

 1. Briefly summarize the content of the manuscript;

The study has described the trends in maternal mortality in Sao Paulo from 2009-2019. The study is important in order to reduce maternal mortality in the region.

2. Illustrate what are, in your opinion, its strengths and weaknesses

(this is an essential step, as the editor will consider the reasoning

behind your recommendation and needs to understand it properly);

However, the authors did not mention mainly the reasons and remedies to cope with the situation. We call this a partially complete study.

3. Provide a point-by-point list of your major recommendations as to

what must be improved.

    The authors need to elaborate more on the causes of maternal mortality (Introduction).

    The authors need to discuss how the trend can be downward (Conclusion).

    The authors need to elaborate more on the methodology section.

    The authors need to brief about methodology and results in the introduction section.

    The authors need to clarify why the recent data for 2020-2023 was not taken in the analysis (Methodology).

    Though the authors discussed some scientific reasons as covariates of maternal mortality, it seems that no solution to these issues was provided.

    APC and AAPC are good calculators but it lacks a number of important aspects. For example, it compares the number only with the previous year's number. Weighting is missing.

    In time series, we prefer moving average. But the authors have used simple averages without giving any reasons.

Author Response

Dear Reviewer 1, 

Below are the changes and responses requested by the reviewer. The article with the changes is attached.

Reviewer 1

Response to Reviewer 1 according to his suggestions and comments.

The introduction part looks incomplete. It is missing in a number of ways. For example, no method was discussed, and no results were briefed.

Second, the reasons for the mortality due to the regions were not researched and not discussed. The research has very weak take-home message for the reader to link it for the actions.

Response: By including in the Abstract section the following excerpt: “This is an ecological study with a time series design, using secondary data from the Mortality Information System (SIM) and the Live Births Information System (SINASC) from 2009 to 2019. All maternal deaths among women aged 10 to 49 years and residing in the state of São Paulo were included in the analysis.” (Lines 22-25).

The leading causes of maternal death were eclampsia (7.13%), gestational hypertension (6.09%), and postpartum hemorrhage (5.89%). (Lines 30-31).

  1. Briefly summarize the content of the manuscript;

The study has described the trends in maternal mortality in Sao Paulo from 2009-2019. The study is important in order to reduce maternal mortality in the region.

Response: The study described the trends in maternal mortality in São Paulo from 2009 to 2019. This study serves as an important foundation for the development of healthcare projects and policies, as it outlines the maternal mortality rate trends over a 19-year period in the most populous state of Brazil, São Paulo. This study can be used as a basis for evaluating the impact of already implemented policies and projects aimed at reducing maternal mortality during the study period, as well as guiding future actions tailored to the specific needs of each region.

  1. Illustrate what are, in your opinion, its strengths and weaknesses (this is an essential step, as the editor will consider the reasoning behind your recommendation and needs to understand it properly);

Response: Strengths:

Highly relevant topic: Maternal mortality is recognized as one of the key global health indicators.

Representative population selection: São Paulo, being the most populous state in Brazil, ensures a representative sample.

Use of reliable secondary data: In Brazil, the completion of the Live Birth Certificate (DNV) and Death Certificate (DO) is mandatory. The government's information systems (SIM and SINASC) are fed by the information contained in these certificates.

Utilization of clusters: The division of São Paulo state into seven regions. For the cluster analysis, the maternal municipality address code was used, geographically delimited by the Administrative Areas established by government agencies and utilized by DATASUS. These clusters include: City of São Paulo - CSP, São Paulo Metropolitan Region excluding the City of São Paulo - RMSP without CSP, São Paulo Metropolitan Region - RMSP, Baixada Santista Metropolitan Region - RMBS, Taubaté Administrative Region - RAT, Central South Cluster - ACS (Sorocaba and Registro Administrative Regions), Campinas and Region Cluster - ACR (Campinas, Piracicaba, and São João da Boa Vista Administrative Regions), Central North Cluster - ACN (Bauru, Araraquara, Ribeirão Preto, and Franca Administrative Regions), Northwest Cluster - ANO (Marília, Presidente Prudente, Araçatuba, São José do Rio Preto, and Barretos Administrative Regions), and State of São Paulo - ESP. This clustering approach ensured an adequate number of cases and stability of the analyses.

Weaknesses:

Own limitation of the study type described in the discussion section.

In-depth analysis of the causality of deaths was not conducted, as the main objective of this study is description-oriented.

However, the authors did not mention mainly the reasons and remedies to cope with the situation. We call this a partially complete study.

Response: This study represents a snapshot focusing on the description of maternal mortality and its trend over a 10-year period from 2009 to 2019 in the state of São Paulo, preceding a global event such as the COVID-19 pandemic. It holds significant relevance for future comparative studies examining earlier and subsequent periods to analyze the impact of policies and the pandemic on this trend.

  1. Provide a point-by-point list of your major recommendations as to what must be improved.

The authors need to elaborate more on the causes of maternal mortality (Introduction).

Response:

Obstetric maternal deaths can be classified as direct or indirect. Direct deaths result from obstetric complications related to pregnancy, childbirth, and the postpartum period, either due to interventions, omissions, incorrect treatment, or a sequence of events resulting from any of these situations (e.g., hemorrhage, puerperal infection, hypertension, thromboembolism, anesthetic accidents). They continue to be the leading cause of maternal mortality in developing countries. Indirect deaths arise from pre-existing or pregnancy-related diseases (intercurrent) that are not directly caused by obstetric factors but are aggravated by the physiological effects of pregnancy (e.g., heart diseases, collagen disorders, and other chronic conditions) (COSTA et al., 2002).

In the descriptive analysis of maternal deaths in the seven clusters of the São Paulo state, the highest number of deaths is concentrated in the Greater São Paulo (GSP) cluster with 1,645 deaths, followed by the Campinas Region Cluster (CRC) with 395 deaths and the Central North Cluster (CNC) with 267 deaths (Table 3).

Regarding the type of obstetric cause of maternal death by clusters, direct causes accounted for the highest number of deaths in all seven regions. Eclampsia (O15) was the most frequent direct cause of death in the following clusters: GSP (121 deaths), CRC (27 deaths), and CNC (22 deaths). Gestational hypertension (O14) was the most frequent direct cause of death in the following clusters: GSP (90 deaths), CRC (27 deaths), and CNC (21 deaths). Postpartum hemorrhage (O72) was the most frequent direct cause of death in the following clusters: GSP (99 deaths), CRC (27 deaths), and CNC (23 deaths) (Table 3).

The most frequent indirect causes in all clusters were: O99 other maternal diseases, classified elsewhere but complicating pregnancy, childbirth, and the postpartum period; O98 other parasitic maternal diseases that complicate pregnancy, childbirth, and the postpartum period; O10 pre-existing hypertension complicating pregnancy, childbirth, and the postpartum period; and O24 diabetes mellitus in pregnancy. Unspecified obstetric causes had a higher number in the GSP and ACR clusters compared to other regions (Table 3).

The authors need to discuss how the trend can be downward (Conclusion)

Response: This is not the objective of this study; however, based on previous studies, the current understanding is as follows:

Qualified care, early identification of pregnant women during prenatal care, referral to high-risk prenatal care, and valuing women's complaints were identified as crucial factors in preventing maternal deaths. In the childbirth assistance, the lack of emphasis on the clinical condition and the absence of connection to a referral hospital that could guarantee a spot and prevent unnecessary travel remain challenges that need to be addressed in the municipality. One key aspect for preventing maternal deaths is the review of cases by Maternal Death Committees, which should adopt an active and participatory approach. All individuals involved in the woman's death should be questioned to identify the factors contributing to the mortality. The case should be examined by qualified professionals committed to education rather than punishment. The objective of the Committee is to analyze all variables involved and propose interventions to prevent future deaths (MACKAY et al., 2005).

A literature review clearly emphasizes the importance of prenatal care for identifying potential risks, ensuring nutritional support for pregnant women, treating diseases, and establishing a maternal immunization program, with the aim of reducing obstetric risks (CALDERON; CECATTI, 2006).

Regarding postpartum care, the lack of follow-up after discharge, delays in diagnosing complications, and early discharge were suggested by family members as factors associated with maternal deaths. Furthermore, strengthening healthcare systems must go hand in hand with social development and equity, both of which are essential steps towards the reduction of preventable maternal deaths (SOUZA et al., 2013).

Calderon, I.M.P.; Cecatti. J.G.; Vega.C.E.P. Intervenções benéficas no pré-natal para prevenção da mortalidade Materna. Rev Bras Gine Obste. v.28, p.310-15, 2006

Souza, J.P.; Gülmezoglu, A.M.; Vogel, J.; Carroli, G.; Lumbiganon, P.; Qureshi, Z.; et al. Moving beyond essential interventions for re- duction of maternal mortality (the WHO Multicountry Survey on Maternal and New- born Health): a cross-sectional study. Lancet. v.381, n.9879, p.1747– 55, 2013.

MacKay A, Berg CJ, Duran C, Chang J, Rosenber H. An assessment of pregnancy-related mortality in the United States. Paediatric and Perinatal Epidemiology., 2005, v.19, pp.206-214.

The authors need to elaborate more on the methodology section.

Response: The study methodology is thoroughly described, encompassing the necessary topics, including study design, study location and period, study population, eligibility criteria, data collection, data analysis, and ethical and legal considerations.

The authors need to brief about methodology and results in the introduction section.

Responsehe authors consider that including this information in the introduction section would be repetitive, as these details have been thoroughly described in the methods and results sections.

The authors need to clarify why the recent data for 2020-2023 was not taken in the analysis (Methodology).

Response: Our intention was precisely to examine a period prior to the Covid-19 pandemic. This study represents a snapshot with a focus on describing maternal mortality and its trend over a 10-year period from 2009 to 2019 in the state of São Paulo, preceding the global event of the Covid-19 pandemic.

It holds significant relevance for future comparative studies analyzing earlier and subsequent periods, aiming to assess the impact of policies and the pandemic on this trend.

Though the authors discussed some scientific reasons as covariates of maternal mortality, it seems that no solution to these issues was provided.

Response: The objective of the study at this moment is not to provide solutions but rather to describe a scenario in order to stimulate further research and provide a foundation for new studies focused on analyzing the measures taken to reduce this indicator.

APC and AAPC are good calculators but it lacks a number of important aspects. For example, it compares the number only with the previous year's number. Weighting is missing.

Response: Using joinpoint regression to identify the time points over the observed period in which a significant change in trend emerged.

In time series, we prefer moving average. But the authors have used simple averages without giving any reasons.

Response: A moving average for daily monitoring is preferable for comparing months within the same year or months between different years, but a simple average can also be used. Moving averages are also useful for decomposing a time series to remove trends or seasonality and make the time series stationary. When the objective is forecasting, other methods for decomposing a time series to achieve stationarity include differencing or regression techniques. It is important to note that this manuscript does not generate forecasts.

Kind regards,

Prof Ribeiro

Reviewer 2 Report

ABSTRACT

The research topic is of scientific and social interest.

There is cohesion in the article, between the section on Theoretical Framework and the subsequent sections, which describe the study and draw conclusions, have a correct, well-structured and cohesive design.

In general, the article is correct and I consider that the topic is in line with the journal’s research objectives.

INTRODUCTION:

The study objective is well defined and identified in both the abstract and the introduction.

The subject under investigation is of growing scientific and social interest. The investigation is current.

The result is a work that can be the basis for many others in this field.

The statistical technique used is well justified and explained, both the process and the results, despite not being widely used in these investigations, which implies an added effort by the authors of the article.

MATERIALS, METHODS and RESULTS:

The statistical treatment is correct.

DISCUSSION and CONCLUSION:

The conclusions are well drawn and interesting.

The discussion is correct.

DEFINITELY:

Therefore, I think it is a good paper and should be accepted for publication in Healthcare.

Author Response

Dear Reviewer 2,

Thank you for encouraging this publication. Thank you for the review and we are glad you enjoyed the writing of this article. 

Kind regards,

Prof. Ribeiro

Reviewer 3 Report

The manuscript entitled “ Describe trends in Maternal Mortality in the State of São Paulo,  Brazil from 2009-2019”: the manuscript needs substantial English language editing by a specified center for language revision, many constructions need revision.

Some other issue are needed to state and revise

-        The first line in the abstract “ The reduction of maternal mortality is a public health problem” how did you describe this reduction as a problem ?

-        “ Mortality Information System and Live Births Information System” : do they have abbreviations to add? , you should mention their address or country.

-        “ the seven regions and São Paulo state” ?  what do you mean by the seven regions , you didn’t present any information about them.

-        Line 44 : how do you start the sentence with “And”?

-        “they adopted a new goal to 53 reduce maternal mortality significantly”, needs revision and rephrasing.

-        Lines 56-58 : need rephrasing and rationale citations.

-        mote prevention and comprehensive and humanized care for these women : and and…???????

-        Is this “ Live Birth Information System” includes a sharp data about the main and actual cause of death to record the cause?

-        You previously mentioned the main causes which lead to maternal deaths , Then in line 87 : you said maternal death between (10  and 49 y ) , is there a mother at 10 years old?

-        Tell me what is the meaning of  “file transfer service”?

-        What is the proper reference you followed to apply your statistics?

-        The cluster map is not clear and needs more explanation under it to explain us why this is presented and show us the cause?

-        I found the aim of the study is not well written.

-        The number of analyzed cases should be mentioned.

-        How is the Maternal Mortality Ratio calculated? Is the result a number or a percent?

-        Page 16 : what does this mean “ omissions” , Add.

-        I found it is better to explain each cause of maternal mortality and its symptoms.

-        “ The direct obstetric cause is the most frequent among types of causes of obstetric 38 death.”  I can’t catch any information from this sentence.

-        How could “ drop in the world fertility ratio” be a public cause in 180 country about maternal deaths?

-        How could the area of study in Brazil could reduce the mortality rate ? tell readers about this policy even in direct or indirect causes?

-        The limitations of the study should be added.

-         

I found the manuscript needs substantial language editing from a specified scientific center.

Author Response

Dear Reviewer 2

Below are the changes and responses requested by the reviewer. The article with the changes is attached.

Reviewer 2

Response to Reviewer 2 according to his suggestions and comments.

The manuscript entitled “Describe trends in Maternal Mortality in the State of São Paulo, Brazil from 2009-2019”: the manuscript needs substantial English language editing by a specified center for language revision, many constructions need revision.

Response: Acknowledged. The revision has been conducted by a native translator/interpreter.

Some other issue are needed to state and revise

  The first line in the abstract “ The reduction of maternal mortality is a public health problem” how did you describe this reduction as a problem ?

Response: We have changed the writing in the text of the abstract. (Line 19)

 “ Mortality Information System and Live Births Information System” : do they have abbreviations to add? , you should mention their address or country.

Response: Incorporating it into the abstract (Lines 23-24) and Methods (Lines 77-87).

“ the seven regions and São Paulo state” ?  what do you mean by the seven regions, you didn’t present any information about them.

Response: In the abstract, it is not feasible to mention all the regions that were divided. In the methods section, the division and the reference used for it are included. The authors' intention was to pique the reader's curiosity to continue reading the full article.

Line 44 : how do you start the sentence with “And”?

Response: Change made (Line 46).

“they adopted a new goal to 53 reduce maternal mortality significantly”, needs revision and rephrasing.

Response: Revised text (Lines 55-58).

Lines 56-58 : need rephrasing and rationale citations.

Response: Revised text (Lines 58-60).

mote prevention and comprehensive and humanized care for these women : and and…???????

Response: Revised text (Lines 64-68).

Is this “Live Birth Information System” includes a sharp data about the main and actual cause of death to record the cause?

Response: Included information explaining about the information system. (Lines 79-87).

You previously mentioned the main causes which lead to maternal deaths, Then in line 87: you said maternal death between (10 and 49 y), is there a mother at 10 years old?

Response: Text included explaining about this age range – Lines 94-99.

 Tell me what is the meaning of “file transfer service”?

Response: The file transfer service is provided by DATASUS (www.datasus.gov.br) through microdata, which has systematic records of epidemiological, morbidity, and vital statistics information in the country and is maintained by the Ministry of Health of Brazil.

What is the proper reference you followed to apply your statistics?

Response:

The cluster map is not clear and needs more explanation under it to explain us why this is presented and show us the cause?

Response: For the same calculation according to territorial clusters, we used the maternal municipality address code as the basis, geographically delimited by the Administrative Areas established by government agencies and used by DATASUS. These clusters were grouped based on similar territorial characteristics (Figure 5). This procedure ensured an adequate number of cases and stability of the analyses.

I found the aim of the study is not well written.

Response: The objective reflects the title of the study, and this is consistent with the rationale of scientific writing.

The number of analyzed cases should be mentioned.

Response: The data were highlighted in the text in the results section.

How is the Maternal Mortality Ratio calculated? Is the result a number or a percent?

Response: The definition can be found in line 136. Maternal mortality is expressed as a number and not a percentage. The calculation is performed per 1000 live births.

Page 16: what does this mean “omissions” , Add.

Response: In line 30 of the discussion section, the term "omissions" has been replaced with "negligent."

I found it is better to explain each cause of maternal mortality and its symptoms.

Response: In the discussion section, lines 38 to 73 provide a synthesis of the main causes in the national and international scenario.

“The direct obstetric cause is the most frequent among types of causes of obstetric 38 death.”  I can’t catch any information from this sentence.

Response: Data in Table 1 informs this frequency.

How could “drop in the world fertility ratio” be a public cause in 180 country about maternal deaths?

Response:

In a systematic review involving 180 countries, the reduction in world maternal mortality stood out, attributed to multiple factors such as the drop in the world fertility ratio, which went from 3.70 in 1980 to 2.56 in 2008, indicating that, despite the growing number of women of reproductive age, the number of births globally remained stable; increase in per capita income; improvement of women's educational level and, finally, qualified care during prenatal care and labor[17].

Reference 17- Ronsmans, C.; Graham, W.J. Maternal mortality: who, when, where, and why. The Lancet, 2006, v.368, pp.1189- 1200.

In this context, it is necessary to consider the changes that have been occurring in the obstetric population profile and maternal mortality, including the reduction in fertility, aging, excessive medicalization, and the increase in chronic degenerative diseases (LEAL et al., 2018).

How could the area of study in Brazil could reduce the mortality rate? tell readers about this policy even in direct or indirect causes?

Response: Studies analyzing mortality rates can serve as a basis for developing more effective projects and public policies. By utilizing data and analysis, it is possible to gain a better understanding of the specific needs of each region's population, thus catering to the regionalization of care.

The limitations of the study should be added.

Response: Lines 145 to 151 of the discussion section.

Kind regards,

Prof Ribeiro

Round 2

Reviewer 3 Report

The limitations and future recommendation of the presented study should be added. 

Neeeds extenssive editing

Author Response

Dear Reviewer 3,

All requested adjustments have been made. What was added is highlighted in yellow.

Kind regards,

Prof. Ribeiro
